# Estimating the Best Time to View Cherry Blossoms Using Time-Series Forecasting Method

**Tomonari Horikawa [1,\*], Munenori Takahashi [1], Masaki Endo [2], Shigeyoshi Ohno [2], Masaharu Hirota [3]** and **Hiroshi Ishikawa [4]**

1   Electronic Information Course, Polytechnic University, Tokyo 187-0035, Japan; m20305@uitec.ac.jp
2   Division of Core, Polytechnic University, Tokyo 187-0035, Japan; endou@uitec.ac.jp (M.E.);
    ohno@uitec.ac.jp (S.O.)
3   Faculty of Informatics, Okayama University of Science, Okayama 700-0005, Japan; hirota@mis.ous.ac.jp
4   Graduate School of Systems Design, Tokyo Metropolitan University, Tokyo 191-0065, Japan;
    ishikawa-hiroshi@tmu.ac.jp
\*   Correspondence: b18321@uitec.ac.jp

**Abstract:** In recent years, tourist information collection using the internet has become common. Tourists are increasingly using internet resources to obtain tourist information. Social network service (SNS) users share tourist information of various kinds. Twitter, one SNS, has been used for many studies. We are pursuing research supporting a method using Twitter to help tourists obtain information: estimates of the best time to view cherry blossoms. Earlier studies have proposed a low-cost moving average method using geotagged tweets related to location information. Geotagged tweets are helpful as social sensors for real-time estimation and for the acquisition of local tourist information because the information can reflect real-world situations. Earlier studies have used weighted moving averages, indicating that a person can estimate the best time to view cherry blossoms in each prefecture. This study proposes a time-series prediction method using SNS data and machine learning as a new method for estimating the best times for viewing for a certain period. Combining the time-series forecasting method and the low-cost moving average method yields an estimate of the best time to view cherry blossoms. This report describes results confirming the usefulness of the proposed method by experimentation with estimation of the best time to view beautiful cherry blossoms in each prefecture and municipality.

**Keywords:** cherry blossom; machine learning; mining; sightseeing; SNS

## 1. Introduction

Today's information-oriented society increasingly presents opportunities for people to obtain information and to use internet resources in daily life. As social networking services (SNSs) become increasingly popular, information of diverse kinds has been distributed and accumulated. In recent years, SNS might be used by tourists to collect tourist information, and to review tourist guidebooks and tourist sites. Sightseeing-related information can then be shared among SNS users by posting from many users. It is not uncommon for tourists to choose from shared tourist information. Twitter [1], which is one example of an SNS that can share tourist information, is generally used to post and browse information such as events and hobbies. Postings of text, photographs, and Twitter images are called "tweets." The user's arbitrary settings can provide the tweet's location information. This tweet is mainly called a "geotagged tweet." Geotagged tweets can share what happened now, and where and what happened. Tweet information from geotagged tweets can then reflect real-world situations. Therefore, it is expected to be helpful as a social sensor for tourists to estimate and acquire local tourist information in real time, irrespective of location. Our research examines the best times to view cherry blossoms using information related to Twitter's geotagged tweets. Earlier studies have proposed a low-cost moving average method

using geotagged tweets with location information. Endo et al. [2] have proposed a simple moving average method. This proposed method can estimate the best time to view cherry blossoms in prefectures and municipalities where a certain number of tweets with geotags are obtainable. Furthermore, the geotagged tweets used for this method are helpful as a social sensor to assess real-world situations. Therefore, this effective estimation of viewing times can provide local tourism information in real time. Takahashi et al. [3] proposed a weighted moving average method (hereinafter "the existing method"). Compared with the simple moving average method, the current method can better calculate the best time to estimate cherry blossoms in each prefecture while maintaining low costs. Earlier reports describe no study of methods for estimating the best time to view cherry blossoms, which is useful for tourist information using AI. Therefore, we describe a systematic method that adopts machine learning analysis after data extraction. This study proposes a time-series prediction method using SNS data with machine learning as a new method for estimating the best time to view cherry blossoms for a certain period. This method uses tweet information accumulated to date as learning data for time-series analysis. Using the machine learning method demonstrates the possibility of automatically predicting the number of tweets in a certain period. Combining the time-series forecasting method and the low-cost moving average method has indicated the best time to view beautiful cherry blossoms. Additionally, we estimate the best time to view cherry blossoms for each prefecture and each municipality. The contribution of this research is an AI-based cherry blossom viewing estimation system. As shown hereinafter, Section 2 introduces the related research. Section 3 describes the experiment method. Section 4 explains the proposed method. Section 5 presents the results. Section 6 presents conclusions.

## 2. Related Research

Great amounts of data that include location information, images, and character strings are accumulated continuously in SNSs. Research efforts have been undertaken to extract such information from SNSs. Moreover, proposals of methods have been conducted. In addition, studies have examined the use of the extracted data for time-series prediction and suggestions for methods. The following describes research on SNSs and time-series analysis.

Hubert et al. [4] surveyed the development of a software tool called Twitter Analytics (TA4GIP). After analyzing sentiment from the correlation of public opinion's reaction to government tweets, we visualized them. Results show that TA4GIP can help identify and analyze aspects of government presence and citizen participation on social media.

Yoshida et al. [5] predicted the tendency of tweets to be retweeted frequently. This trend was investigated using two methods: potentially retweeted scoring (PWRTS) and reverse retweet propensity scoring (IRTS). As a result, the IRTS trend for tweet detection indicates that it might be worth consideration in both English and Japanese.

Amati et al. [6] conducted a temporal analysis of the Twitter stream to investigate the evolution of unique events based on bursts of the popularity of their associated hashtags. They derived a classification of events according to different patterns corresponding to the peak of the volume of exchanged messages and the propagation of these events on social networks having characteristics that are identical to those of Twitter.

These studies [4–6] are similar to this study in that event analysis is performed after information is extracted from Twitter. However, this study extracts location information and text information from Twitter. The best time to see the cherry blossoms is estimated.

Yang et al. [7] described TimeSeries AggregatoR (TSAR)—a robust, scalable, real-time event time-series aggregation of a framework built primarily for engagement monitoring. It performs aggregation of interactions with Tweets and segmentation along with a multitude of dimensions such as devices and engagement types. Built on Summingbird, TSAR was designed to manage the examination and processing of events to publish results in heterogeneous data stores. The query interface powers dashboards and supports downstream ad hoc analytics provided for clients.

Chen et al. [8] propose a time series generative model used for nonparametric online time series classification called a latent source model. It establishes a theoretical performance guarantee for weighted majority voting under a latent source model and predicts which news topics on Twitter will go viral to become trends.

These studies [7,8] are similar to this study in that they perform time series predictions for events. However, in this study, we used AI to predict the number of tweets about cherry blossoms in time series.

Leelaprute et al. [9] proposed a method for extracting travel records from check-in information stored in the SNS shared location information: "Foursquare". Then they built a database of travel records. Additionally, they proposed a method for selecting travel records that match individual tastes and interests from a database. Consequently, obtaining a sufficient number of travel records to support the creation of tour plans has become possible.

Morishita et al. [10] proposed "SakuraSensor", which uses an in-vehicle smartphone to extract landscape route information automatically from videos taken. SakuraSensor, as a participatory sensing system, shares data among users nearly in real time. It has confirmed the flowering state of cherry blossoms. It can identify the state with a recall rate of 84% and precision of about 74%. Additionally, it confirmed that the k-stage sensing method can obtain the same POI detection rate in 1/2 of the detection time of the conventional method.

Zang et al. [11] simulate climate and phenological suitability for botanical tourism over the next 30 years, based on climate data from different scenarios in Beijing. It was predicted that the spring botanical tourism period from 2040 to 2050 would be early. The autumn botanical tourism period is expected to be later than that in the past 55 years.

These studies [9–11] are similar to the present study in that they conduct experiments using tourism information. However, for this study, we propose a cherry blossom viewing system that uses Amazon Web Services (AWS) [12].

As described above, research assessing SNSs and time-series analysis has been conducted, but no report of the relevant literature describes a study using the methods to estimate the best time to view cherry blossoms. Therefore, this study was conducted using time-series prediction to estimate the best certain period for viewing them.

## 3. Experiment Method

This chapter presents descriptions of preprocessing and the data used. Using the Streaming API [13], we collected geotagged tweets with location information, including latitude and longitude. Then we also analyzed each tweet sent by a prefecture or municipality using the National Institute of Agriculture, Forestry, and Fisheries' simple reverse geocoding service [14] from geotagged tweets' latitude and longitude information, including location information. The planning area analyzed general towns and streets within the named city. Additionally, we analyzed tweets with biological names. From 1 February 2015 to 29 July 2019, we explored the transition of tweets related to organism names of three types: "さくら" (Hiragana), "サクラ" (Katakana), and "桜" (Chinese characters). As this study's best time estimation method, we used the existing method, which uses a weighted moving average. Earlier studies demonstrated that the peak of cherry blossoms can be predicted by combining tweet information and best-time estimation conditions. The existing method is explained below.

### 3.1. Weighted Moving Average

The weighted moving average used for the existing method is a moving average with each value assigned a weight. Adding weights improves the recall and accuracy of best time estimation. For the existing method, the median is set to 1. Values $\pm$ 0.5 from the median respectively represent minimum and maximum values. In addition, except for the median, weights from the lowest value to the highest value are assigned linearly. The value is rounded to the third decimal place. For example, taking the 5-day weighted moving average used for the existing method, the following Equation (1) is obtained. Here, $H_{avg5}$

represents the weighted moving average for 5 days; $x_y$ denotes the number of tweets $x$ before $y$ days prior.

$$H_{avg5} = (X_5 \times 0.5 + X_4 \times 0.75 + X_3 \times 1 + X_2 \times 1.25 + X_1 \times 1.5)/5. \tag{1}$$

*3.2. Best Time Estimation Method*

We used a simple moving average and a weighted moving average. The following estimation method was set for the frequency of appearance of each geotagged tweet including the target word. Results were analyzed by date to estimate the best viewing period. 1. We used a one-year simple moving average to ascertain the period during which tweets about cherry blossoms increased; 2. Tweets tend to be more numerous on Saturdays and Sundays. Therefore, the 7-day weighted moving average was used weekly; 3. A 5-day weighted moving average was used based on the average number of days from the flowering of cherry blossoms to full bloom: 5 days. Using these best time estimation criteria, the best time to view blossoms at each tourist spot was inferred.

## 4. Proposed Method

We propose machine learning as a new method for estimating the best time to view cherry blossoms. In the existing method, the best time is estimated by the application of conditions of the best time (1-year moving average, 5-day weighted moving average, 7-day weighted moving average, etc.) to the number of tweets. Therefore, by predicting the number of tweets, it might be possible to estimate the best time to see a set period. This study was conducted to assess a new method for estimating the best time to see cherry blossoms after a certain period. The number of tweets about cherry blossoms from 1 February 2015 to the day before the estimated best time to see them was used in the existing method. The best time for viewing was estimated every time this tweet information is updated daily. The estimated best time to view the cherry blossoms is the two months from 1 March to 30 April. Therefore, using the conventional method, the best time to see the cherry blossoms is estimated daily during the two months of the best time to view cherry blossoms.

Using the proposed method, the number of tweets was predicted in time series using tweet information about cherry blossoms accumulated from 1 February 2015, as learning data. The time-series forecast period was two months from 1 March to 30 April 2019. Subsequently, the best time to view cherry blossoms was estimated every two months using the number of tweets predicted in chronological order. Therefore, the proposed method can collectively estimate the optimum time for viewing in units of two months instead of the optimum time for estimation in units of one day, as in the conventional method. We used AWS. Time-series forecasting was performed using the Amazon Forecast [15] function of AWS. The learning data of this study included seasonal data such as the number of data for 300 days or more and the number of tweets about cherry blossoms. Therefore, the Amazon Forecast algorithm used DeepAR+ [16], optimal for large-scale learning data of 300 days or more, and Prophet [17], optimal for seasonal learning data. The time-series prediction method by DeepAR+ is proposed method (1). The time-series prediction method by Prophet is proposed method (2). For the predicted value, a value that satisfies the demand of 50% obtained using the weighted quantile loss was used. Weighted quantile loss is a metric for forecasting using Amazon Forecast. The predicted number of tweets was rounded down to the nearest whole number. In addition, when the predicted number of tweets became negative, it was treated as 0 tweets. The condition of the best time to estimate the cherry blossoms was the same as with the existing method. The use of machine learning will verify whether the number of tweets in a certain period can be predicted automatically. Figure 1 portrays a schematic diagram of this study.

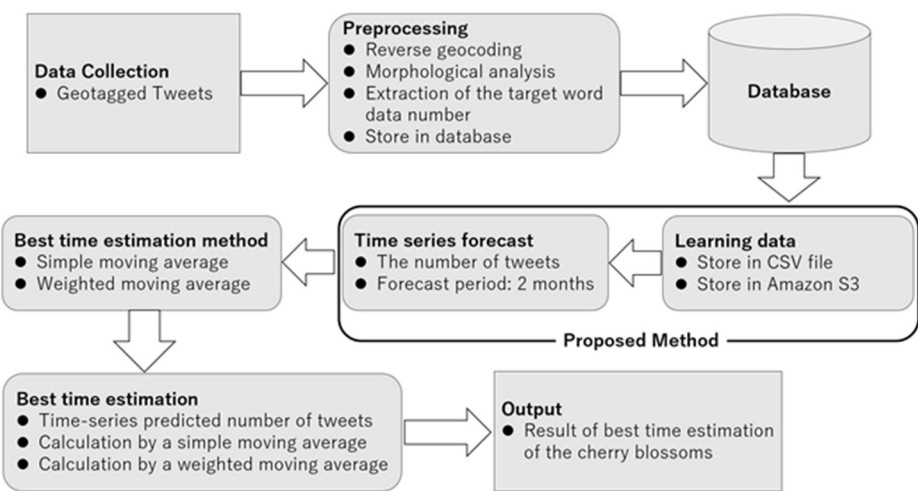

**Figure 1.** Schematic diagram.

*4.1. Proposed Method (1)*

The tweet information accumulated from 1 February 2015 (the number of tweets about cherry blossoms for the date of each prefecture) was used as learning data. The target areas for the experiment were Tokyo, Kyoto, and Shizuoka. The alphabet that identifies the prefecture, the date (Format: yyyy-mm-dd), and the number of tweets were stored in the CSV file in order from the left. After storage, the learning data were saved in Amazon S3 [18] on AWS. First, time-series forecasts from 1 March to 30 April 2019 were conducted using tweet information from 1 February 2015 to 28 February 2019 in the experiment target area as learning data. Although tweets about Sakura usually appear frequently from early February through late April, it was difficult to predict the number of tweets in time series. That is true because, in units of one year, the characteristic period is short, rendering it as unsuitable for the learning data of proposed method (1). Therefore, the learning data in the target area were set from 1 February through 30 April from 2015–2018 and from 1 February through 28 February 2019. The period including characteristic data was extracted. Because the actual date became a discontinuous value when the characteristic period was extracted, the data were converted so that the time-series prediction period became a constant value. The time-series forecast period for tweets was two months from 1 March to 30 April 2019.

*4.2. Proposed Method (2)*

The tweet information accumulated from 1 February 2015 (the number of tweets about cherry blossoms for the date of each prefecture) was used as learning data. The target areas for the experiment were Tokyo, Kyoto, and Shizuoka. The alphabet that identified the prefecture, the date, and the number of tweets were stored in the CSV file in order from the left. After storage, the learning data were saved in Amazon S3 on AWS. Unlike proposed method (1), the learning data of proposed method (2) were the tweet information from 1 February 2015 to 28 February 2019 in the target area. The time-series forecast period for tweets was two months from 1 March to 30 April 2019.

*4.3. Estimation for Each Municipality*

The number of tweets predicted in time series using proposed method (2) tended to be higher than the actual number of tweets. This fact might be useful for best time estimation in each prefecture and in each municipality. Therefore, the best time for viewing is estimated for each municipality using proposed method (2). The target areas for the experiment were the Meguro ward, Chiyoda ward, and Hachioji city. The preconditions for time-series forecasting for each municipality in this study are described. We thought that we could make a more accurate prediction by aligning the position information of the tweet information, which was the learning data, to some extent. Therefore, we will incorporate

the tweet information of the municipality and the prefecture of that municipality into the learning data. When estimating the best time to see the Meguro ward, we would use the tweet information of Tokyo and the tweet information of the Meguro ward.

### 4.4. Proposed Method (2) in 2021

To reinforce the conclusions, we describe the estimation of the best time to see cherry blossoms using proposed method (2) in years other than 2019. For this study, we will conduct an experiment in 2021 for each prefecture. We did not acquire tweet information in 2020 because of the convenience of the system. Therefore, the tweet information for 2020 will not be considered. The target areas for the experiment were Tokyo, Kyoto, and Shizuoka. The learning data in the target area were set from 1 February 2015 to 31 December 2019 and from 1 January to 28 February 2021. The time-series forecast period for tweets was two months from 1 March to 30 April 2021.

## 5. Results

This chapter presents the results of the best time to view cherry blossoms, as estimated using the proposed method. From earlier studies, the peak of cherry blossoms can be predicted by combining tweet information and best-time estimation conditions. The accuracy of peak prediction is evaluated by comparing the best time estimation result and the correct answer data, and by the values of recall and precision. The correct answer data use the cherry blossom date to full bloom date observed by the Japan Meteorological Agency. The cherry blossom day is the first day when 5–6 or more cherry buds are open on the sample tree. The full bloom day of the cherry tree indicates the first day on which more than 80% of the buds of the sample tree are open. Therefore, the best time to view blossoms will continue even after the full bloom date. Consequently, the precision rate might decrease. As comparison targets of the proposed method, Figures 2–4 show the best times to view cherry blossoms in Tokyo, Kyoto, and Shizuoka prefectures in 2019, as inferred using the existing method.

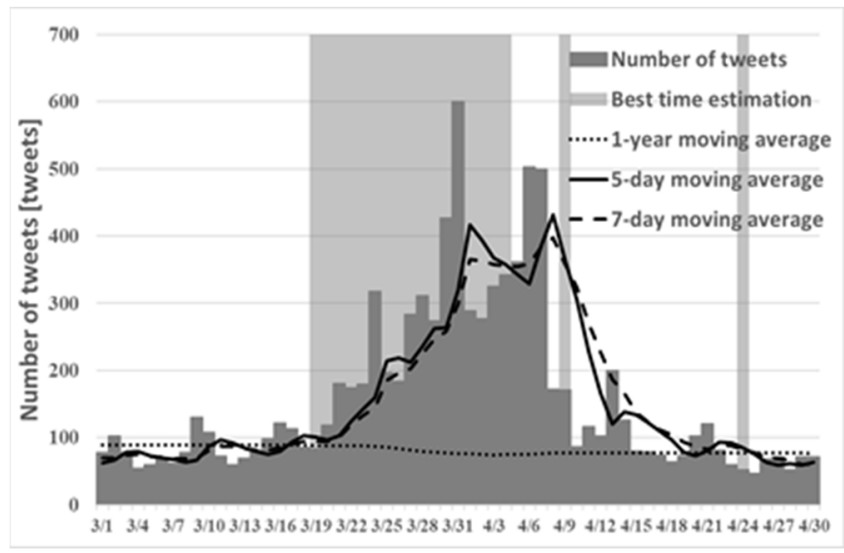

**Figure 2.** Tokyo in 2019.

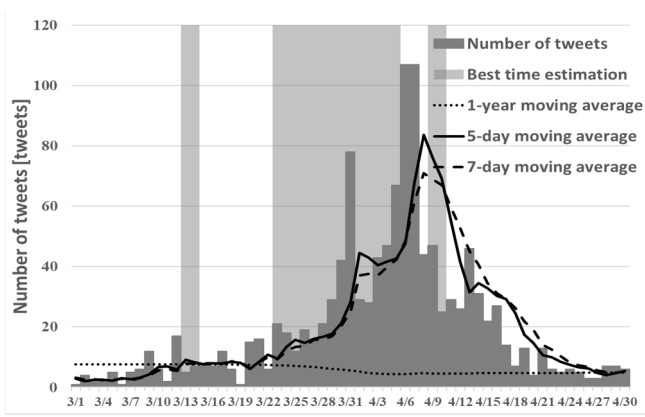

**Figure 3.** Kyoto in 2019.

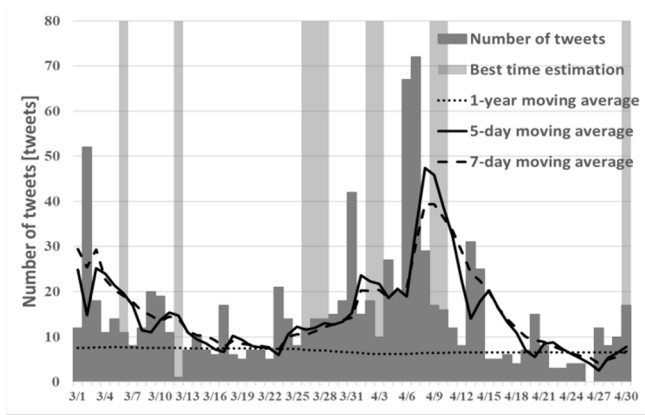

**Figure 4.** Shizuoka in 2019.

### 5.1. Results of Proposed Method (1)

Using proposed method (1) to estimate the condition of the cherry blossoms with the conventional method, we describe the number of tweets related to the cherry blossoms predicted in time series and the best time estimation result obtained for the best time. Figures 5–7 shows the best times to see cherry blossoms in Tokyo, Kyoto, and Shizuoka prefectures in 2019 using proposed method (1). Table 1 presents a comparison of the actual total number of tweets from 1 March to 30 April 2019 and the total number of tweets predicted in chronological order by proposed method (1). Table 2 presents comparison of the recall rate and precision rate of the best time estimation using the existing method and best time estimation using proposed method (1).

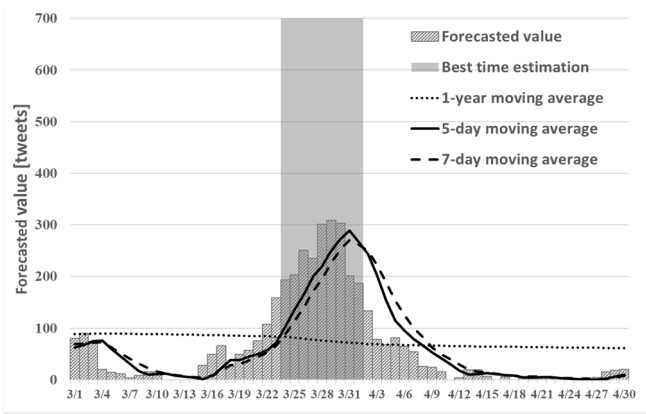

**Figure 5.** Tokyo after forecasting in 2019 (proposed method (1)).

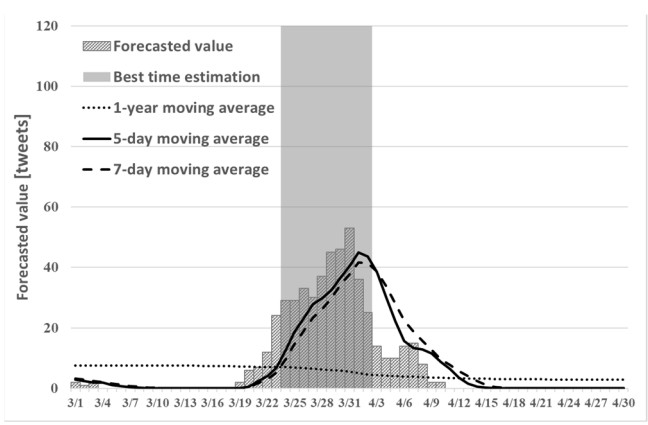

**Figure 6.** Kyoto after forecasting in 2019 (proposed method (1)).

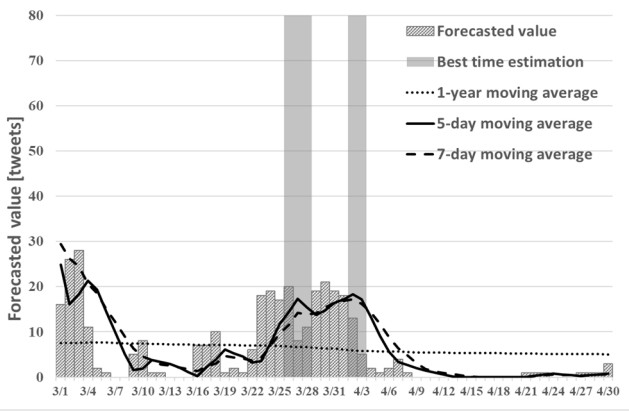

**Figure 7.** Shizuoka after forecasting in 2019 (proposed method (1)).

**Table 1.** Number of tweets, actual total, and the forecasted total (proposed method (1)).

|  | Area | Number of Tweets |
|---|---|---|
| Actual | Tokyo | 9586 |
| Forecasted | | 3810 |
| Actual | Kyoto | 1226 |
| Forecasted | | 494 |
| Actual | Shizuoka | 902 |
| Forecasted | | 340 |

**Table 2.** Recall rate and precision rate between existing method and proposed method (1).

|  | Area | Recall | Precision |
|---|---|---|---|
| Existing method | Tokyo | 100.0% | 36.8% |
| Proposed method (1) | | 57.1% | 33.3% |
| Existing method | Kyoto | 90.9% | 52.6% |
| Proposed method (1) | | 63.6% | 50.0% |
| Existing method | Shizuoka | 30.0% | 17.6% |
| Proposed method (1) | | 30.0% | 25.0% |

Comparison of Figures 2–4 and Figures 5–7 can confirm prediction of the number of tweets. Particularly in the figure for Shizuoka in Figure 7, it can confirm that the peak is in mid-March. However, the cherry blossoms at the beginning of March are presented because early blooming cherry blossoms (early February through early March), such as "Kawazu cherry blossoms" and "Kakegawa cherry blossoms", are famous in Shizuoka. Therefore, using proposed method (1), one can predict the tendency that is peculiar to a

region such as Shizuoka. However, from Table 1, the number of tweets predicted in time series using proposed method (1) tended to be smaller than the actual number of tweets. In addition, from Table 2, the recall rate of proposed method (1) in Tokyo and Kyoto was lower than that obtained using the existing method.

### 5.2. Results of Proposed Method (2)

Using proposed method (2), we describe the number of tweets related to the cherry blossoms predicted in time series and the best time estimation result obtained using the best time to estimate the condition of cherry blossoms of the conventional method. Figures 8–10 show the best time to see cherry blossoms in Tokyo, Kyoto, and Shizuoka prefectures in 2019 using proposed method (2). Table 3 presents a comparison of the actual total number of tweets from 1 March to 30 April 2019 and the total number of tweets predicted in chronological order using proposed method (2). Table 4 presents a comparison of the recall rate and precision rate of the best time estimation by the existing method and the best time estimation using proposed method (2).

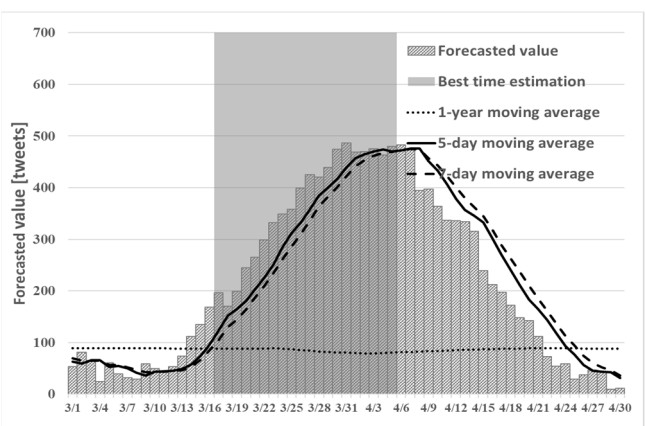

**Figure 8.** Tokyo after forecasting in 2019 (proposed method (2)).

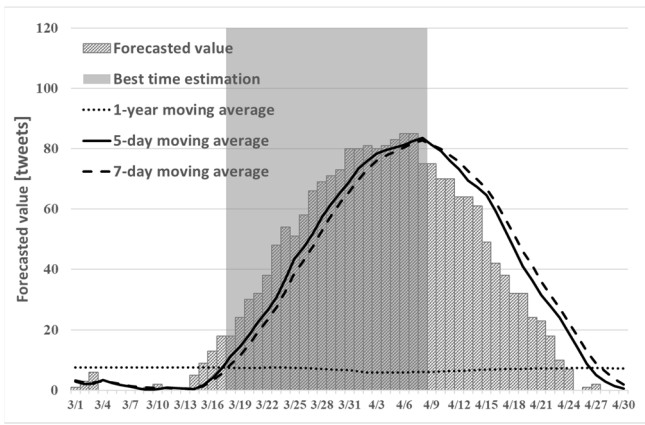

**Figure 9.** Kyoto after forecasting in 2019 (proposed method (2)).

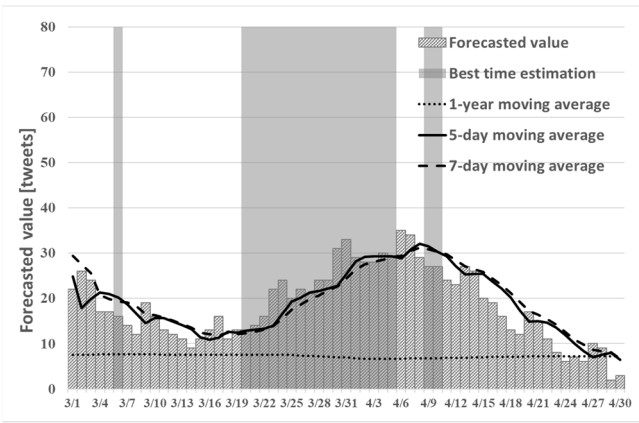

**Figure 10.** Shizuoka after the forecast in 2019 (proposed method (2)).

**Table 3.** Number of tweets, actual total and the forecasted total (proposed method (2)).

| | Area | Number of Tweets |
|---|---|---|
| Actual | Tokyo | 9586 |
| Forecasted | | 13,510 |
| Actual | Kyoto | 1226 |
| Forecasted | | 2102 |
| Actual | Shizuoka | 902 |
| Forecasted | | 1126 |

**Table 4.** Recall rate and precision rate between existing method and proposed method (2).

| | Area | Recall | Precision |
|---|---|---|---|
| Existing method | Tokyo | 100.0% | 36.8% |
| Proposed method (2) | | 100.0% | 35.0% |
| Existing method | Kyoto | 90.9% | 52.6% |
| Proposed method (2) | | 100.0% | 50.0% |
| Existing method | Shizuoka | 30.0% | 17.6% |
| Proposed method (2) | | 90.0% | 42.9% |

A comparison of Figures 2–4 and Figures 8–10 can confirm the prediction of the number of tweets. Particularly in the figure for Shizuoka in Figure 10, it can confirm that the peak is in mid-March. However, the cherry blossoms at the beginning of March are presented because early blooming cherry blossoms (early February through early March), such as "Kawazu cherry blossoms" and "Kakegawa cherry blossoms", are famous in Shizuoka. Therefore, using proposed method (2), one can predict the tendency that is peculiar to a region such as Shizuoka. From Table 3, the number of tweets predicted in time series using proposed method (2) tended to be larger than the actual number of tweets. In addition, from Table 4, the recall rate of proposed method (2) in Kyoto and Shizuoka was higher than that obtained using the existing method.

*5.3. Results of Estimation for Each Municipality*

For each municipality, we describe the number of tweets related to the cherry blossoms predicted in time series and the best time estimation result obtained using the best time to estimate the condition of the cherry blossoms of the conventional method. In each municipality, Table 5 presents a comparison between the actual total number of tweets from 1 March to 30 April 2019 and the total number of tweets predicted in chronological order using proposed method (2). In each municipality, Table 6 presents a comparison of the recall rate and precision rate of the best time estimation by the existing method and the best time estimation using proposed method (2).

**Table 5.** For each municipality, number of tweets, the actual total and the forecasted total.

|            | Area    | Number of Tweets |
|------------|---------|------------------|
| Actual     | Meguro  | 930              |
| Forecasted |         | 1607             |
| Actual     | Chiyoda | 1630             |
| Forecasted |         | 1950             |
| Actual     | Hachioji| 128              |
| Forecasted |         | 121              |

**Table 6.** For each municipality, the recall rate and precision rate.

|                     | Area     | Recall  | Precision |
|---------------------|----------|---------|-----------|
| Existing method     | Meguro   | 100.0%  | 41.2%     |
| Proposed method (2) |          | 100.0%  | 31.8%     |
| Existing method     | Chiyoda  | 57.1%   | 28.6%     |
| Proposed method (2) |          | 100.0%  | 38.9%     |
| Existing method     | Hachioji | 28.6%   | 8.7%      |
| Proposed method (2) |          | 100.0%  | 31.8%     |

As shown in Table 5, in the Meguro and Chiyoda wards, the number of tweets predicted in time series using proposed method (2) tended to be higher than the actual number of tweets. However, in Hachioji city, the number of tweets predicted in time series using proposed method (2) tended to be lower than the actual number of tweets. From Table 6, the recall rate of proposed method (2) was higher than that of the existing method in Chiyoda ward and Hachioji city.

### 5.4. Results of Proposed Method (2) in 2021

Using proposed method (2), in 2021, we describe the number of tweets related to the cherry blossoms predicted in time series and the best time estimation result obtained using the best time to estimate the condition of cherry blossoms of the conventional method. Figures 11–13 show the best time to see cherry blossoms in Tokyo, Kyoto, and Shizuoka prefectures in 2021 using proposed method (2). Table 7 presents a comparison of the actual total number of tweets from 1 March to 30 April 2021 and the total number of tweets predicted in chronological order using proposed method (2). Table 8 presents a comparison of the recall rate and precision rate of the best time estimation by the existing method and the best time estimation using proposed method (2), in 2021.

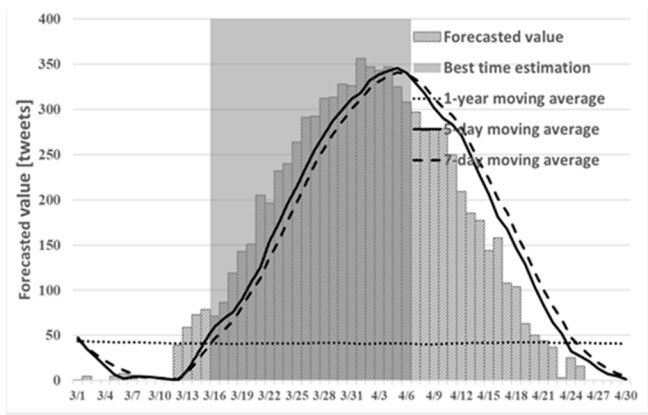

**Figure 11.** Tokyo after forecasting in 2021 (proposed method (2)).

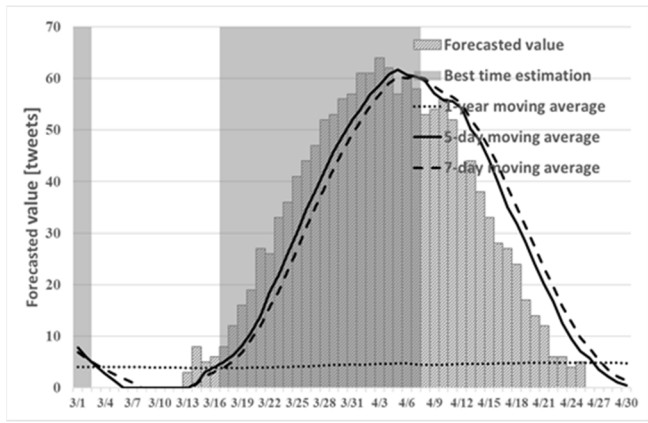

**Figure 12.** Kyoto after forecasting in 2021 (proposed method (2)).

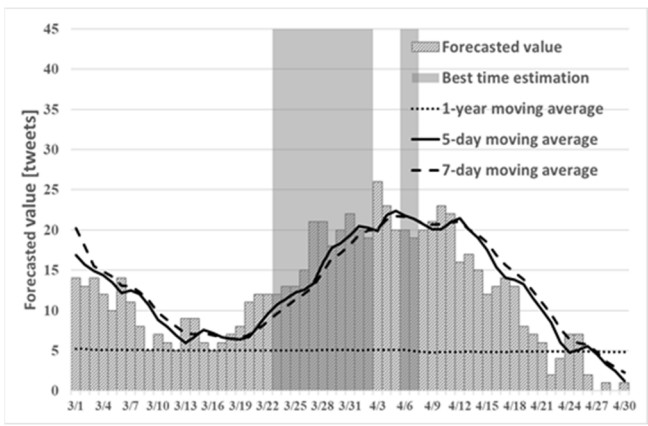

**Figure 13.** Shizuoka after the forecast in 2021 (proposed method (2)).

**Table 7.** Number of tweets in 2021, actual total and the forecasted total (proposed method (2)).

|  | Area | Number of Tweets |
|---|---|---|
| Actual | Tokyo | 4289 |
| Forecasted |  | 8572 |
| Actual | Kyoto | 665 |
| Forecasted |  | 1487 |
| Actual | Shizuoka | 671 |
| Forecasted |  | 737 |

**Table 8.** Recall rate and precision rate between existing method and proposed method (2), in 2021.

|  | Area | Recall | Precision |
|---|---|---|---|
| Existing method | Tokyo | 55.6% | 25.0% |
| Proposed method (2) |  | 77.8% | 29.2% |
| Existing method | Kyoto | 63.6% | 30.4% |
| Proposed method (2) |  | 90.9% | 40.0% |
| Existing method | Shizuoka | 22.2% | 12.5% |
| Proposed method (2) |  | 66.7% | 37.5% |

Figures 1–13 can confirm the prediction of the number of tweets. Particularly for Shizuoka in Figure 13, it can be confirmed that the peak is in mid-March. However, the cherry blossoms at the beginning of March are presented because early blooming cherry blossoms (early February through early March), such as "Kawazu cherry blossoms" and "Kakegawa cherry blossoms", are famous in Shizuoka. Therefore, using proposed method

(2), it is possible to detect regional peaks not only in 2019 but also in 2021. From Table 7, in 2021, the number of tweets predicted in time series using proposed method (2) tended to be larger than the actual number of tweets. In addition, from Table 8, in 2021, the recall rate of proposed method (2) in the experiment target area was higher than that obtained using the existing method.

## 6. Conclusions

This article has described experimentation with a new method of estimating the best time to view cherry blossoms for a certain period. We proposed a time-series prediction method using SNS data using machine learning and verified it through experimentation using proposed method (1) and proposed method (2).

Results obtained using proposed method (1) confirmed that the time-series forecasting of the number of tweets can predict a tendency peculiar to the region. However, in the experiment target area, the tweets predicted in time series using proposed method (1) were fewer than the actual number of tweets, which affected the best time estimation result. We compared the recall rates obtained using the existing method and proposed method (1). The recall rate in Tokyo decreased from 100.0% to 57.1%. The recall rate in Kyoto decreased from 90.9% to 63.6%. The recall rate was maintained in Shizuoka at 30.0%. Therefore, compared with the recall rate of the existing method, the method using proposed method (1) was able to retain the recall rate of Shizuoka. However, it was not possible to improve the recall rate of Tokyo or Kyoto.

Proposed method (2) confirmed that the time-series forecasting of the number of tweets can predict a tendency peculiar to the region. In addition, in the experiment target area, the number of tweets predicted in time series using proposed method (2) was higher than the actual number of tweets, which positively affected the best time estimation result. We compared the recall rate from the existing method and proposed method (2). Proposed method (2) maintained the recall rate in Tokyo at 100.0%. The recall rate in Kyoto improved from 90.9% to 100.0%. The recall rate in Shizuoka improved from 30.0% to 90.0%. Therefore, compared with the recall rate obtained using the existing method, proposed method (1) was able to maintain the recall rate of Tokyo. Additionally, it was possible to improve the recall rate in Kyoto and Shizuoka.

We can describe the best time to estimate each municipality using proposed method (2). For each municipality, we compared the recall rate by the existing method and proposed method (2). It maintained the recall rate in the Meguro ward at 100.0%. The recall rate in the Chiyoda ward improved from 57.1% to 100.0%. Then the recall rate in Hachioji city improved from 28.6% to 100.0%. Therefore, compared with the recall rate of the existing method, the method using proposed method (2) in each municipality was able to maintain the recall rate of the Meguro ward. Additionally, it was possible to improve the recall rate in the Chiyoda ward and Hachioji city. However, in Hachioji city, the tweets predicted in time series using proposed method (2) tended to be fewer than the actual number of tweets. For time series forecasting for each municipality, further study of the learning data is necessary.

In addition, to confirm the effectiveness of the proposed method, the best time to estimate results obtained using proposed method (2) in 2021 will be described. Using proposed method (2), it was possible to detect regional peaks not only in 2019 but also in 2021. It is possible to compare the recall rate using the existing method and the recall rate by proposed method (2) in 2021. The recall rate in Tokyo improved from 55.6% to 77.8%. The recall rate in Kyoto improved from 63.6% to 90.9%. The recall rate in Shizuoka improved from 22.2% to 66.7%. Therefore, in 2021, compared with the recall rate obtained using the existing method, the method using proposed method (2) was able to improve the recall rate in the experiment target area.

Based on the discussion presented above, we confirmed that the time-series prediction of the number of tweets can predict the tendency peculiar to the region. We demonstrated the possibility of estimating the best time for a certain period ahead using the time-series

forecasting method. As a challenge for future study, we intend to proceed with verification by increasing target items and target areas, which will realize a time-series prediction system using SNS data that combines seasonal data.

**Author Contributions:** Investigation, T.H.; Supervision, M.T., M.E., S.O., M.H. and H.I.; Validation, T.H.; Writing—original draft, T.H. All authors have read and agreed to the published version of the manuscript.

**Funding:** This research was supported by JSPS KAKENHI (I.H.) grant number (20K12081) and (H.M.) grant number (19K20418) and (E.M.) grant number (18K13254), and the Okawa Foundation Research Grant.

**Institutional Review Board Statement:** Not applicable.

**Informed Consent Statement:** Not applicable.

**Data Availability Statement:** Not applicable.

**Conflicts of Interest:** The authors declare no conflict of interest.

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
