# Peer review of "Estimating the Best Time to View Cherry Blossoms Using Time-Series Forecasting Method"

_make, doi:10.3390/make4020018_

Round 1

Reviewer 1 Report

Estimating the best time using time-series forecasting method 

The article attempts to describe an approach to predict the best time to watch cherry blossoms in different areas of Japan by analyzing the trends in number of related tweets. The forecast accuracy is compared with another similar approach that uses a slightly different forecast equation.

While the problem that is being studied is an interesting application of social media analysis, the article should be improved. Specifically:

It is not clear what is the main contribution. Is it the novelty of the application, a new forecast equation, a new way of getting relevant tweets?

The description of the overall approach is not clear. It will help to include a diagram of the various components of the prediction system. Some things that are not clearly described are how is the peak of the cherry blossom predicted (from the predicted peak of the SNS messages peak?), how is precision/recall defined for this problem.

Many of the works cited in the related work do not seem relevant for the work described in the paper. It is better to cite work that uses social media analysis for other tourist applications, or which uses peaks of messages (e.g., [6]). It is also important to relate the cited work to the proposed approach.

All the evaluation is done on the 2019 season. Conclusions will be stronger if the system performs similarly in other years too. For instance, can the region-specific peaks ("Kawazu bloom") be detected in other years? It should be possible to use some portion of the training data for evaluation; also 2020 data is available.

The title of the paper should be changed to be more descriptive. As the system is specific for cherry blossom bloom prediction, this should be part of the title.

Reviewer 3 Report

This is an interesting comparative analysis of time series forecasting methods. The paper is reasonably well contextualised and it has been applied to an interesting case. However, the case itself raises some questions. The authors have applied the comparison of methods to a particular seasonal phenomena. Given the major challenge that seasonality provides tourism it would be appropriate to have examined different forms of seasonal phenomena. However, even with what you have examined I am very surprised that there has been no attempt to observe the relationship to related natural phenomena such as observed weather data, which could have potentially made the forecasting even more powerful. Although it is not necessary to do this it would be worthwhile to at least provide more commentary on both seasonality in tourism and potential relationship to other data sources that could be used for predictive purposes.

Round 2

Reviewer 1 Report

The authors have adequately addressed the comments from the previous review. The overview diagram is helpful. In particular, the results from 2021 are even better. The readability of the paper could still be improved. For instance, when is the detected peak considered accurate? Is it "when the peak falls within the range of cherry blossom date to full bloom date from the Japan Meteorological Agency"? How wide is this range typically?